# Outcomes of Esophageal Cancer after Esophagectomy in the Era of Early Injection Laryngoplasty

**DOI:** 10.3390/diagnostics11050914

**Published:** 2021-05-20

**Authors:** Tuan-Jen Fang, Yu-Cheng Pei, Yi-An Lu, Hsiu-Feng Chung, Hui-Chen Chiang, Hsueh-Yu Li, Alice M. K. Wong

**Affiliations:** 1Department of Otolaryngology Head & Neck Surgery, Chang Gung Memorial Hospital at Linkou, 5 Fushing St., Taoyuan 333, Taiwan; b9402009@cgmh.org.tw (Y.-A.L.); s77001birdphone@yahoo.com.tw (H.-F.C.); hyli38@cgmh.org.tw (H.-Y.L.); 2School of Medicine, Chang Gung University, 259 Wen-Hwa 1st Road, Taoyuan 333, Taiwan; yspeii@adm.cgmh.org.tw (Y.-C.P.); walice@cgmh.org.tw (A.M.K.W.); 3Department of Physical Medicine and Rehabilitation, Chang Gung Memorial Hospital at Linkou and Taoyuan, 5 Fushing St., Taoyuan 333, Taiwan; 4Center for Vascularized Composite Allotransplantation, Chang Gung Memorial Hospital, 5 Fushing St., Taoyuan 333, Taiwan; 5Healthy Aging Research Center, Chang Gung University, 259 Wen-Hwa 1st Road, Taoyuan 333, Taiwan; 6Graduate School of Management, Ming Chun University, 250 Zhong Shan N Rd., Sec.5, Taipei 111, Taiwan; hcchiang@mail.mcu.edu.tw

**Keywords:** unilateral vocal fold paralysis, esophagectomy, esophageal cancer, injection laryngoplasty, survival

## Abstract

(1) Background: severe weight loss was reported to be related to unilateral vocal fold paralysis (UVFP) after esophagectomy and could thus impair survival. Concomitant radical lymph node dissection along the recurrent laryngeal nerve during esophageal cancer surgery is controversial, as it might induce UVFP. Early intervention for esophagectomy-related UVFP by administering intracordal injections of temporal agents has recently become popular. This study investigated the survival outcomes of esophagectomy for esophageal squamous cell carcinoma (ESCC) after the introduction of early injection laryngoplasty (EIL). (2) Methods: a retrospective review of patients with ESCC after curative-intent esophagectomy was conducted in a tertiary referral medical center. The necessity of EIL with hyaluronic acid was comprehensively discussed for all symptomatic UVFP patients. The survival outcomes and related risk factors of ESCC were evaluated. (3) Results: among the cohort of 358 consecutive patients who underwent esophagectomy for ESCC, 42 (11.7%) showed postsurgical UVFP. Twenty-nine of them received office-based EIL. After EIL, the glottal gap area, maximum phonation time and voice outcome survey showed significant improvement at one, three and six months measurements. The number of lymph nodes in the resected specimen was higher in those with UVFP than in those without UVFP (30.1 ± 15.7 vs. 24.6 ± 12.7, *p* = 0.011). The Kaplan–Meier overall survival was significantly better in patients who had UVFP (*p* = 0.014), received neck anastomosis (*p* = 0.004), underwent endoscopic resection (*p* < 0.001) and had early-stage cancer (*p* < 0.001). Multivariate Cox logistic regression analysis showed two independent predictors of OS, showing that the primary stage and anastomosis type are the two independent predictors of OS. (4) Conclusion: EIL is effective in improving UVFP-related symptoms, thus providing compensatory and palliative measures to ensure the patient’s postsurgical quality of life. The emerging use of EIL might encourage cancer surgeons to radically dissect lymph nodes along the recurrent laryngeal nerve, thus changing the survival trend.

## 1. Introduction

The incidence of recurrent laryngeal nerve (RLN) injury caused by surgery for esophageal squamous cell carcinoma (ESCC) can vary from 10% to 63% [1,2,3,4]. Dissection of the lymph nodes along the RLN could eradicate metastatic cancer but increase the incidence of vocal fold paralysis [5]. Although spontaneous recovery is expected to occur in some of these patients [4], the related nutrition and pulmonary complications [1,6,7] still impact the survival outcomes, making radical lymph node dissection controversial.

The period of observation required after esophagectomy-related unilateral vocal fold paralysis (UVFP) and its treatment remains inconclusive. Conventional treatments of UVFP, such as thyroplasty and autologous fat injection, are recommended after the patient’s vocal dysfunction remains impaired after the wait-and-see period. This period is always suggested to be 9–12 months after symptom occurrence. Patients might suffer from persistent dysphonia, aspiration, and impaired pulmonary clearance from ineffective cough during this period of observation. Under some circumstances, tube feeding or tracheostomy might be necessary for nutrition and to prevent aspiration pneumonia [8,9,10]. Therefore, these potential consequences may make cancer surgeons hesitate to perform radical lymph node dissections.

Early injection laryngoplasty (EIL) is the procedure of injecting the temporary material into the paralyzed vocal fold to correct the vocal position. The voice and swallowing function can be restored immediately after the procedure. It has been reported to improve quality of life in patients with UVFP after thoracic surgery [11]. The long-term clinical outcomes of esophageal cancer surgery may also be altered by the introduction of such novel management principles. We suggest that EIL performed by laryngologists can change the cancer management strategy determined by thoracic surgeons if the outcome of EIL is promising. The aims of the study are to evaluate the risk factors for UVFP after esophageal cancer surgery and the impact of UVFP on survival after the introduction of office-based EIL.

## 2. Materials and Methods

The study was approved by the Institutional Review Board of Chang Gung Memorial Foundation (201305333A3, 24 February 2013). From October 2010 to July 2016, 358 consecutive patients who underwent esophagectomy for esophageal squamous cell carcinoma (ESCC) were recruited. The inclusion criteria were as follows: (1) age over 18 years old, (2) curative-intent treatment, and (3) complete pre-treatment staging process. The preoperative vocal fold status could be a sign of tumor extension, so that, in cases with preoperative UVFP should be carefully evaluated carefully to determine the tumor status. In this cohort, the patients which were operable with a suspected clear surgical margin were recruited. None of them had UVFP before surgery. The general characteristics, operation records, and clinical and pathologic reports were reviewed. All UVFP patients underwent voice laboratory analysis and quality of life measurements, and patients who received EIL with hyaluronate were evaluated again after one month.

### 2.1. Principle of Diagnosing and Managing UVFP

Following esophagectomy, all patients were evaluated by the laryngologist (TJF) if dysphonia or aspiration was observed post-operatively. Videolaryngostroboscopy was used to assess vocal fold motility. Neurogenic UVFP was diagnosed if immobile vocal folds were found with videolaryngostroboscopy, and denervation changes in the vocal fold adductors were identified using laryngeal electromyography.

Starting in July 2010, the technique of office-based EIL was introduced to our institute and changed the esophageal cancer management protocol. Early intervention, which is performed within 6 months of esophagectomy for symptomatic UVFP patients, by EIL using hyaluronate, became the standard of care. Office-based EIL was suggested for patients with severe breathy voice or intractable aspiration.

### 2.2. Office-Based Injection Laryngoplasty

Patients undergoing the procedure were seated upright on a semi-reclined chair without any sedation. Vital signs, including blood pressure and heartbeat, were monitored before the injection and every 10 min during the injection. The nose and throat were numbed with a 10% lidocaine spray. Then, the assistant inserted a fiberoptic laryngoscope with a distal chip (Laryngoscope: ENF TYPE V2; Platform: EVIS Exera II; Olympus Optical Co, Ltd., Tokyo, Japan) into the patient’s nostril to observe the glottal area. After the larynx was exposed, approximately 0.5–1.0 mL 2% lidocaine was injected in the subcutaneous layer over the cricothyroid (CT) membrane.

An injection needle was passed through the CT membrane on the paramidline point. After penetrating the CT membrane, the surgeon moved the laryngoscope slightly upwards submucosally, and confirmed the tip location by moving it medially. When the site of injection was confirmed, up to 1 mL hyaluronate was injected into the vocal fold just lateral to the vocal ligament. During the procedure, the patient was asked to produce voice until a satisfactory sound was achieved.

### 2.3. Assessments of Patients with Unilateral Vocal Fold Paralysis

After esophagectomy, each patient’s voice and swallowing function were evaluated one week post-operatively. Those with hoarseness or swallowing impairment were referred to the laryngologist for a laryngoscopic examination. If unilateral vocal fold paralysis was suspected, the patients were sent for a complete primary and follow-up assessment. The specific assessment included laryngeal electromyography (LEMG), videolaryngostroboscopy, a voice outcome survey (VOS), laboratory voice analysis, and an SF-36 health-related quality of life survey.

### 2.4. LEMG Examination

LEMG was performed when vocal fold movement impairment was noted during laryngoscopy. The protocol of LEMG has been reported in a previous publication [12]. By using a concentric needle electrode, electric signals were obtained. While recording the TA-LCA complex activity, the patient was asked to produce three series of “i”s at three different intensities (low, moderate, and highest possible), with each “i" lasting at least 400 ms [12]. In CT function testing, a glissando upward “i" at normal loudness was produced three times. The standard technique was reported in our previous publication [13].

To quantify the LEMG measurements, the recorded raw LEMG data were analyzed by an automatic algorithm to localize the timing and amplitude of each turn [12]. Turn frequency was defined as the turn numbers divided by the duration of each epoch. Turn frequencies of the TA-LCA muscle complex were averaged for epochs, and the peak turn frequency was obtained from the turn frequencies among the top three epochs.

### 2.5. Glottal Position: Normalized Glottal Gap Area

When performing videolaryngoscopy examinations, the patients were asked to produce “i" at a modal pitch and regular loudness. The glottal conformations were recorded for several phonatory cycles. The normalized glottal gap area (NGGA), defined as the glottal gap area/normal vocal fold length squared [14], was analyzed using image processing software (ImageJ 1.44p, National Institutes of Health, Bethesda, MD, USA).

### 2.6. Laboratory Voice Analysis

Computerized software (Computerized Speech Lab model 4300B, version 5.05; Kay Elemetrics Corp., Lincoln Park, IL, USA) was utilized to analyze the voice sample with a sampling rate of 25.6 kHz and 16-bit quantization. The acoustic data, including fundamental frequency, jitter (perturbation of frequency), shimmer (perturbation of amplitude), and harmonic-to-noise ratio, were tabulated from the recorded voice. The longest duration of a sustained vowel “a” determined the maximum phonation time.

### 2.7. Quality of Life: Voice Outcome Survey (VOS) and Short-Form 36 (SF-36)

The VOS, originally developed by Gliklich et al. in the 1990s, is a specific UVFP-related quality of life assessment. It is based on a five-point Likert scale and evaluates the physical and social problems associated with UVFP [15]. The validated Mandarin Chinese version was applied in this project [16]. Health-related quality of life was evaluated using the SF-36 questionnaire. The International Quality of Life Assessment Standard Taiwan, version 1.0, was applied in this study [17,18].

### 2.8. Statistical Analysis

The data were analyzed using SPSS (SPSS Inc., Chicago, IL, USA). Fisher’s exact test and Student’s t test were used to compare categorical and parametric data between UVFP and non-UVFP patients, respectively. Comparisons of the baseline assessments between UVFP patients with and without immediate treatment were performed using Student’s t-tests for parametric data. Comparisons of the changes in parameters over time after EIL injection were conducted using repeated-measures ANOVA. Univariate analyses of the risk of survival according to clinicopathological factors were performed using the log-rank test. Both known clinical and pathological risk factors for postoperative outcomes, such as clinical stage, existence of UVFP, esophageal anastomosis, and individual risk factors selected by univariate analyses, were included in the model. The survival curves were generated using the Kaplan–Meier method. To examine overall survival according to clinical factors, the log-rank test was used. The α value was defined as 0.05. Multivariate analysis of survival was conducted using a stepwise logistic regression model fitted using a backward selection procedure, by including variables with *p* values < 0.05 in univariate analysis.

## 3. Results

During the study period, 358 (339 males and 19 females, aged 55.4 ± 9.4 years) consecutive patients who underwent curative-intent esophagectomy for primary ESCC were recruited. At the initial examination, most had advanced-stage disease: early-stage disease (stages I and II): 30.4%, and advanced stage (stages III and IV): 69.6%. Approximately 11.7% of the study cohort or 42 patients who had voice or swallowing impairment were diagnosed with UVPF, and most of them (31 over 42 or 73.8%) had left UVFP. After a median follow-up of 26 ± 23 months, 4 patients recovered from UVFP (Table 1).

The patients with UVFP had significantly more dissected lymph nodes than those without vocal fold paralysis (30.1 ± 15.7 vs. 24.6 ± 12.7; *p* = 0.011). Minimally invasive esophagectomy (thoracoscopy) and neck anastomosis are two risk factors for UVFP. Specifically, UVFP is more likely to occur after thoracoscopy than after open chest anastomosis (17.5% vs. 1.2%; *p* < 0.001) and is more likely to occur after neck anastomosis than after chest anastomosis (16.1% vs. 2.6%; *p* < 0.001). There were no differences in age (*p* = 0.673) or stage (*p* = 0.469) between the patients with and without UVFP.

The 5-year overall survival (OS) rate of this patient cohort was 36%. The OS rate of patients without UVFP was 34%, which was significantly lower than that of patients with UVFP (60%, *p* = 0.014). The patients with UVFP also had better disease-free survival than those without UVFP, but the difference was not significant. (Figure 1B)

The results showed that the status of vocal cord motion as well as the anastomosis site, esophagectomy technique and primary cancer stage affect OS. Specifically, better OS was observed in the groups who received neck anastomosis, had early-stage disease, and underwent minimally invasive esophagectomy (Figure 2). Using multivariate Cox logistic regression analysis, the primary stage and esophagectomy technique were identified as independent predictors of OS (Table 2).

Twenty-nine (70.7%) UVFP patients received early office-based EIL, and the interval from esophagectomy to EIL was 3.3 ± 1.8 months. The number of EIL procedures and the intervals from esophagectomy to EIL are shown in Figure 3.

Compared to the no EIL group, the EIL group had a worse clinical presentation in the glottal area during laryngoscopy, voice analysis data, turn ratio in laryngeal electromyography, and VOS score, as shown in Table 3. Similarly, according to the SF-36 questionnaire, the EIL group had worse quality of life regarding physical functioning (*p* = 0.003) and role limitations due to emotional problems (*p* = 0.014) than the no EIL group. For patients who underwent EIL, most parameters improved significantly one, three and six months after EIL. (Table 4)

## 4. Discussion

Patients with vocal paralysis after esophagectomy were reported to have poorer outcomes than those without vocal paralysis. The resulting breathy voice and aspiration of liquid impair quality of life and increase the risk of pneumonia. UVFP after esophagectomy was also reported to increase the risk of severe weight loss and pulmonary consequences [4,6,10]. Previous reports claimed that the risk factors include lymphadenectomy around the right recurrent laryngeal nerve (RLN), cervical esophagus mobilization and cervical anastomosis [4,19,20,21,22]. In this study, we noted that patients with neck anastomosis who underwent minimally invasive endoscopic esophagectomy were more prone to developing UVFP. Although some of the UVFP patients eventually recovered through compensation or reinnervation, post-operative care remains a great challenge. Routine jejunostomy tube feeding or prophylactic tracheostomy has been reported to prevent such consequences [10,23,24], but quality of life is substantially impaired with these management methods.

The RLNs run along the tracheoesophageal groove. During the dissection process, the RLN can be injured by electrocauterization, stretching or compression from the surrounding edema. When performing anastomosis, the cervical esophagus is usually pulled upwards from the left side of the neck [1]. Compared to open thoracotomy, the relatively narrow dissection field of the thoracoscope may increase the difficulty in identifying the RLN. Thus, more cases of UVFP were observed after minimally invasive endoscopic esophagectomy with cervical anastomosis in this study.

Although cervical anastomosis leads to a higher incidence of UVFP, thoracic surgeons might still choose this approach for reconstruction as it has some benefits. The potential advantages of cervical anastomosis over thoracic anastomosis include a lower chance of local recurrence, lower mortality if leakage occurs, and less need to expand the thoracic incision. Although recovery from the endoscopic procedure and cervical anastomosis is more rapid than that from open thoracotomy, when vocal fold paralysis occurs, subsequent weight loss and pulmonary complications further impact long-term outcomes [6,25]. The high incidence of post-operative UVFP [4] and its consequences may cause some surgeons to avoid such minimally invasive procedures, neck anastomoses, or radical lymph node dissection along the RLN during esophagectomy.

In this study, open thoracotomy and advanced-stage disease were the only two risk factors related to overall survival. In contrast to previous reports [1,4,25], our results showed that patients with UVFP have better overall and disease-specific survival rates than those without nerve injuries. However, after multivariate analysis, its role on survival was diminished. We noted patients of thoracoscopy with cervical anastomosis had a higher rate of UVFP. The above two factors and numbers of dissected node may also decrease the role of UVFP impaction on OS. The fact that UVFP did not have worse outcomes can be attributed to the introduction of EIL. From 2011–2016, the interval between EIL and esophagectomy shortened in our institute (Figure 3). All esophagectomy patients with symptomatic UVFP received EIL within three months in 2016. Although temporary side effects from UVFP may still exist, after immediate UVFP treatment, the symptoms can be corrected, reducing the influence of UVFP on survival.

Office-based injection laryngoplasty for UVFP has been reported for decades. The procedure was popularized after the development of a novel distal chip scope and new fillers such as hyaluronic acid [26,27]. From our previous report [11], the patients’ voice and quality of life immediately improved after office-based intracordal injection of hyaluronic acid. Most of the patients experienced less aspiration during deglutition after EIL. Such changes further encouraged the patients to eat more to restore their nutritional status. Starting in 2010, the routine evaluation and early intervention for post-operative UVFP reduced the incidence of aspiration and restored swallowing function [11]. Early injection laryngoplasty has also been reported to reduce the necessity for intensive care [28,29]. From our results, the injection procedure did not interrupt adjuvant therapy and did not lead to any complications. Immediate restoration of the patients’ voice and swallowing function were noted starting from the first measurement. Therefore, adjuvant therapy for esophageal malignancies would not be delayed.

The survival outcomes of cancer surgery have been reported to be related to the grade of lymph node dissection. In this study, more lymph nodes were noted in the dissected specimen of patients with UVFP than in those of patients without UVFP. In the past, the relatively high morbidity after UVFP prevented the surgeon from aggressively dissecting along the RLN. However, the high success rate of immediate UVFP management through office-based injection laryngoplasty may further encourage thoracic surgeons to dissect more aggressively and thus improve disease control and survival.

There are still some limitations in this present study. First, the study period was long, and the criteria for selecting open thoracotomy or minimally invasive esophagectomy may change over time. However, by adjusting for confounding factors, the risks of open thoracotomy remained robust. Second, patients with asymptomatic UVFP might be ignored. All incidences of UVFP were diagnosed from the symptoms of breathy voice and aspiration. Without such symptoms, patients might not be referred to laryngologists and may be excluded from the study. A prospective cohort study can reduce the bias and further investigate the influence of these factors.

## 5. Conclusions

Patients who underwent minimally invasive endoscopic esophagectomy and received neck anastomosis were more likely to be complicated with UVFP. Office-based injection laryngoplasty immediately improves the quality of life of patients with UVFP. After applying the principle of early intervention for UVFP, the survival rate of patients with UVFP is not inferior to those with intact vocal functions.

## Figures and Tables

**Figure 1 diagnostics-11-00914-f001:**
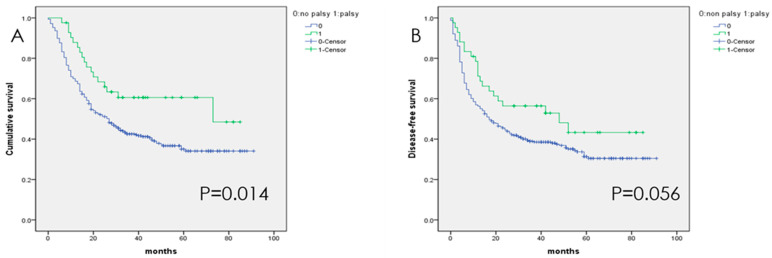
Overall (**A**) and disease-free (**B**) survival of patients after esophagectomy. Patients with unilateral vocal fold paralysis had a significantly better survival rate than those without vocal fold paralysis.

**Figure 2 diagnostics-11-00914-f002:**
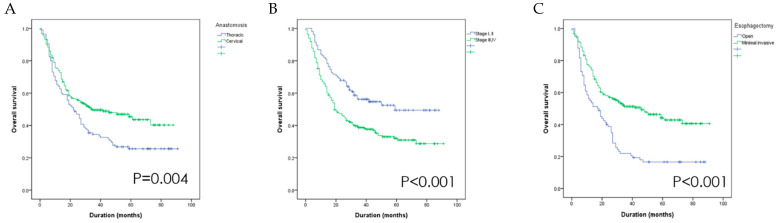
The overall survival is better in groups that received neck anastomosis (**A**), had stage I or II disease (**B**) and underwent minimally invasive esophagectomy (**C**).

**Figure 3 diagnostics-11-00914-f003:**
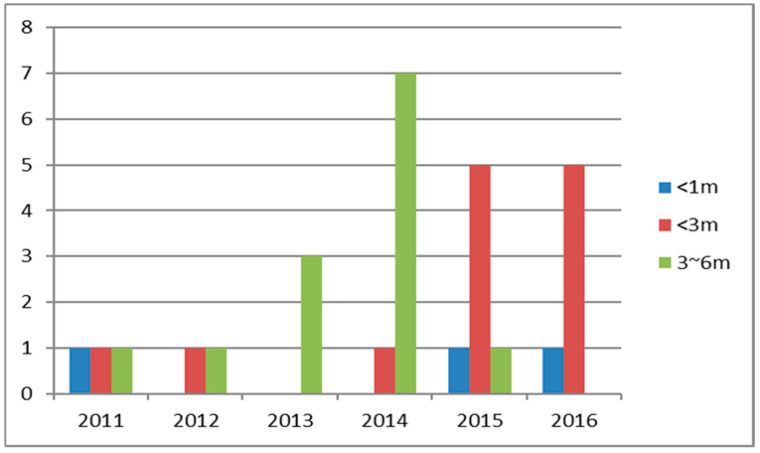
From 2011 to 2016, the number of injection laryngoplasty procedures for unilateral vocal fold paralysis has increased, and this procedure is now performed earlier.

**Table 1 diagnostics-11-00914-t001:** Patient characteristics of the enrolled group.

	All	Non-UVFP *	UVFP	*p*
Sex (male/female)	339/19	300/16	39/3	0.459
Age (years)	55.42 ± 9.41	55.50 ± 9.61	54.63 ± 7.39	0.665
Paralysis		316	42	
Side (left/right)			36/6	
Hyaluronate injection (Y/N)			31/11	
Recovery (Y/N)			4/38	
Thoracic anastomosis/cervical anastomosis	115/241	112/204	3/39	0.000
Thoracotomy/MIE **	82/274	81/235	1/41	
Stage	11	9	3	
IA				
IB	23	21	2	
IIA	20	16	4	
IIB	54	48	6	
IIIA	90	80	80	
IIIB	79	70	70	
IIIC	63	57	57	
IV	14	14	0	
Unknown	2	1	1	
Alive/dead	147/209	123/193	25/17	

* UVFP: unilateral vocal fold paralysis; ** MIE: minimally invasive esophagectomy.

**Table 2 diagnostics-11-00914-t002:** Multivariate analysis of the factors influencing overall survival after esophagectomy.

	HR		95% CI	*p*
Vocal fold motility	Palsy	1	0.889–2.454	0.133
	Intact	1.48		
Surgical procedure	MIE	1	1.408–3.072	<0.001
	Open	2.08		
Stage	I, II	1	1.243–2.375	0.001
	III, IV	1.72		
Anastomosis	Cervical	0.91	0.626–1.311	0.600
	Thoracic	1		

**Table 3 diagnostics-11-00914-t003:** Comparison of the characteristics of patients with UVFP who did or did not undergo immediate injection laryngoplasty.

Parameters	*n* = 29	*n* = 13	
	HA Injection	No HA Injection	*p* Value
Glottal gap			
Closed-phase NGGA	11.67 ± 12.80	3.11 ± 4.09	0.037 *
	20.55 ± 12.93		0.072
	8.87 ± 6.21		0.705
LEMG			
Normal side of TA-LCA (turn/s)	1112.92 ± 410.48	904.39 ± 277.32	0.131
Lesion side of TA-LCA (turn/s)	297.20 ± 227.69	582.73 ± 634.41	0.173
Turn ratio of TA-LCA	0.33 ± 0.31	0.69 ± 0.72	0.034 *
	914.05 ± 346.93		0.531
	704.05 ± 301.44		0.814
	0.83 ± 0.38		0.510
Voice laboratory analysis			
Maximum phonation time (s)	3.93 ± 2.84	10.83 ± 7.34	0.016 *
SZ ratio	2.05 ± 1.03	1.41 ± 0.93	0.097
Fundamental frequency (Hz)	156.97 ± 51.97	131.62 ± 37.71	0.197
Jitter (%)	5.18 ± 5.35	1.71 ± 1.12	0.051
Shimmer (dB)	0.98 ± 0.65	0.48 ± 0.29	0.024 *
Harmonic-to-noise ratio	5.15 ± 2.99	7.37 ± 5.79	0.128
Vos			
Voice outcome survey score	30.71 ± 12.15	58.50 ± 24.04	0.005 **
SF-36			
Physical functioning	60.36 ± 25.27	80.50 ± 13.22	0.003 **
Role limitation due to physical health	16.96 ± 34.06	30.00 ± 43.78	0.342
Role limitation due to emotional problem	30.92 ± 40.50	70.00 ± 42.89	0.014 *
Vitality	47.50 ± 21.54	54.50 ± 11.65	0.337
Mental health	63.29 ± 18.60	64.80 ± 13.96	0.816
Social functioning	47.45 ± 25.51	65.00 ± 26.87	0.074
Bodily pain	63.61 ± 24.91	74.75 ± 25.51	0.235
General health perceptions	41.79 ± 22.53	55.00 ± 19.72	0.110

* *p* < 0.05, ** *p* < 0.01.

**Table 4 diagnostics-11-00914-t004:** The change of parameters in the comprehensive voice assessment after HIL.

Parameters	Baseline (a)	1 Month Post HIL (b)	3 Month Post HIL (c)	6 Month Post HIL (d)	*p* Value	Significant Comparison ^†^
Glottal gap	*n* =31	*n* =28	*n* =25	*n* =20		
Closed-phase NGGA	11.3751 ± 12.4384	2.0742 ± 2.1528	4.844 ± 4.598	2.9502 ± 4.2794	0.009 **	ab, ac, ad
Voice laboratory analysis						
Maximum phonation time (s)	3.8067 ± 2.7767	7.5948 ± 6.1624	7.2396 ± 6.0317	7.5043 ± 5.9946	<0.001 ***	ab, ac, ad
SZ ratio	2.1704 ± 1.1316	1.4734 ± 0.9986	1.6349 ± 1.0143	1.6355 ± 1.2799	0.072	ab
Fundamental frequency (Hz)	154.8151 ± 51.1644	142.7253 ± 32.9603	144.229 ± 34.116	146.8968 ± 38.6546	0.774	
Jitter (%)	5.1929 ± 5.1651	1.988 ± 1.043	2.8703 ± 1.9116	2.2739 ± 1.5287	0.050	ab
Shimmer (dB)	1.008 ± 0.6403	0.4184 ± 0.17	0.5679 ± 0.3546	0.6688 ± 0.7645	0.042 *	ab, ac
Harmonic-to-noise ratio	4.9816 ± 3.0341	7.3429 ± 2.6551	6.3831 ± 1.8776	7.2882 ± 2.5396	0.004 **	ab, ac, ad
Voice outcome survey						
Score of voice outcome survey	30.3448 ± 12.0957	58.0357 ± 18.7251	55 ± 21.7945	62.1053 ± 25.6209	<0.001 ***	ab, ac, ad
SF-36						
Physical functioning	60.83 ± 24.88	66.03 ± 25.05	71.54 ± 20.29	68.75 ± 25.64	0.053	ab, ac
Role limitation due to physical health problems	15.83 ± 33.14	26.72 ± 38.92	36.54 ± 44.85	41.25 ± 48.17	0.053	ac, ad
Role limitation due to emotional problems	28.86 ± 39.86	48.23 ± 43.28	51.22 ± 44.48	50 ± 51.3	0.034 *	ac
Vitality	47.67 ± 20.79	51.03 ± 17.44	51.73 ± 19.23	57.75 ± 21.12	0.001 **	ac, ad
Mental health	62.13 ± 18.49	66.34 ± 13.07	66.62 ± 15.76	66 ± 19.1	0.107	ac
Social functioning	46.38 ± 25	61.76 ± 26.94	66.98 ± 26.2	71.25 ± 24.03	<0.001 ***	ab, ac, ad
Bodily pain	62.72 ± 24.72	80.02 ± 22.82	77.71 ± 21.22	79.28 ± 21.59	0.003 **	ab, ac, ad
General health perceptions	41.33 ± 23.19	49.83 ± 20.55	45.96 ± 20.4	50.75 ± 20.41	0.360	

*: *p* < 0.05; **: *p* < 0.01; ***: *p* < 0.001; ^†^: significant comparison pairs; ab: between baseline and 1 month post HIL; ac: between baseline and 3 month post HIL; ad: between baseline and 6 month post HIL. NGGA: normalized glottal gap area.

## Data Availability

Data available on request due to restrictions, e.g., privacy or ethical. The data presented in this study are available on request from the corresponding author. The data are not publicly available due to patients’ privacy.

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
