# Peer review of "Outcomes of Esophageal Cancer after Esophagectomy in the Era of Early Injection Laryngoplasty"

_diagnostics, 2021, doi:10.3390/diagnostics11050914_

Round 1

Reviewer 1 Report

The Authors report on the outcome results of early injection laryngoplasty in order to treat unilateral vocal fold paralysis in oesophageal cancer patients undergoing curative-intent oesophagectomy with lymphnode dissection along the recurrent laryngeal nerve. The topic is of interest, since post-surgical swallowing disorders may increase the likelihood for aspiration pneumonitis consequently jeopardizing patient’s global outcome. I do have comments:

  • I miss the data of baseline UVFP. The presence of UVFP at diagnosis (before oesophagectomy), due to tumor compression and/or infiltration, could be mirrorin more aggressive disease or more advanced stage (beyond TNM classification), with a potential correlation with the final prognosis. UVFP patients could have a worst prognosis, because of UVFP at baseline, due to more aggressive disease. I would suggest the authors to provide details on this.
  • Can you please provide details on the histologies of the tumor in the present cohort? I assume most of the cases would be SCC, as it is typical in Asia, but a reporting on the histology distribution would be helpful to interpret the survival data.
  • Did any patient undergo combined modality treatment, including pre-operative or post-operative radiation? I would expect it to be likely since the relative proportion of locally advanced cases is high. Could you please provide details on this, since radiation may affect the functionality of the laryngeal nerve and consequently the motility of the vocal cords. I would compare the 2 groups with respect to the variable radiotherapy.
  • Any CDDP-based chemotherapy. I would exploit the same consideration as per radiotherapy. I would compare the 2 groups with respect to the variable chemotherapy.
  • In your cohort, OS for patients with vocal cord paralysis is higher than OS for those without. This finding is counterintuitive and goes in the opposite direction of what you discussed in the introduction. Why is that, in your opinion? Which are the confounders?

Author Response

Dear esteemed editor and reviewers,

    We are very grateful of the comments given by the editor and reviewers. The point-to-point correspondence to each comment is listed as follows. We hope the revised manuscript is suitable for publication in Diagnostics.

Sincerely yours,

Tuan-Jen Fang, MD

Chang Gung Memorial Hospital, No. 5 Fushing St., Taoyuan 333, Taiwan

Tel.: +(886)-3328-1200 ext. 3846; Fax: +(886)-3328-1200 ext. 2667;

E-mail: fang3109@cgmh.org.tw

Reviewer 1

I miss the data of baseline UVFP. The presence of UVFP at diagnosis (before oesophagectomy), due to tumor compression and/or infiltration, could be mirrorin more aggressive disease or more advanced stage (beyond TNM classification), with a potential correlation with the final prognosis. UVFP patients could have a worst prognosis, because of UVFP at baseline, due to more aggressive disease. I would suggest the authors to provide details on this.

Thanks for bring this up to us. We added the following passage in the Materials and Methods section, as:

“The patients which is operable with suspected clear surgical margin were recruited. None of them had UVFP before surgery.”

Since we recruited only the cases accepted curative-intent surgery, those with preoperative far-advanced stage cases were not recruited.

Can you please provide details on the histologies of the tumor in the present cohort? I assume most of the cases would be SCC, as it is typical in Asia, but a reporting on the histology distribution would be helpful to interpret the survival data.

Thanks for the comment. The manuscript focused on esophageal “Squamous cell carcinoma” which we described in the line 5 of the “Materials and methods”. Those with other histology types were not included.

Did any patient undergo combined modality treatment, including pre-operative or post-operative radiation? I would expect it to be likely since the relative proportion of locally advanced cases is high. Could you please provide details on this, since radiation may affect the functionality of the laryngeal nerve and consequently the motility of the vocal cords. I would compare the 2 groups with respect to the variable radiotherapy. Any CDDP-based chemotherapy. I would exploit the same consideration as per radiotherapy. I would compare the 2 groups with respect to the variable chemotherapy.

The treatment guideline with respect to the preoperative status was standardized.

Specifically, for early stages disease (stages I-II), surgical resection is the primary option for those without contraindications; for advanced stages disease (stages III-IV), neo-adjuvant therapy with platinum-based chemotherapy (PF mostly) and 3-4000cGy radiation were performed before surgery. Then wide excision would be performed in surgical candidates. Because the treatment plan was standardized and were not tailored according to the patient’s pre- or postoperative laryngeal nerve status, we did not include it in the results.

To tackle this issue, we added the following passages in Materials and methods section, as”

“The preoperative vocal fold status could be a sign of tumor extension, so that  preoperative UVFP should be carefully evaluated to determine the tumor status.” Future studies are needed to characterize the degree to which preoperative vocal fold status affects tumor outcomes

In your cohort, OS for patients with vocal cord paralysis is higher than OS for those without. This finding is counterintuitive and goes in the opposite direction of what you discussed in the introduction. Why is that, in your opinion? Which are the confounders?

Thanks for the question. We suggest EIL is the answer for better survive in UVFP. Although aggressive dissection along the esophagus would achieve a wider margin and furthermore, a better oncologic outcome, this approach is also more prone to cause UVFP. Before the era of EIL, patients with UVFP usually suffers from the pneumonia that may impair survival. After the introduction of EIL, it can immediate correct UVFP symptoms and decrease UVPD-induced dysphagia and thus to improve OS.

The issue has been discussed in the 5th paragraph of “Discussion” section.

Reviewer 2 Report

Congratulations on this study on a large cohort of patients with esophageal SCC undergoing esophagectomy and, partially, EIL.

I have a few suggestions on how the manuscript could be improved.

  • In the abstract, you should also present results supporting your conclusion that "EIL is effective in improving UVFP-related symptoms,". So far, there are none.
  • In the abstract, you will also need to state that the survival advantage for patients with UVFP is found only in univariate analysis, and vanishes when doing a multivariable analysis. The way you present results now is misleading.
  • In the introduction, you should already provide a brief description of what EIL actually is, so readers unfamiliar with the technique can get a first idea.
  • In the multivariable regression analysis, I suggest defining stage I/II as reference category (OR=1), and stage III/IV as comparator. This would make ORs across possible risk factors more easily comparable. In the methods section, you fail to accurarely describe how you construct and calculate the multivariable model.
  • The fact that the effect of UVFP on survival loses significance after multivariable adjustment reflects the fact that there is selection who gets radical lymph node dissection along the RLN.
  • Figure 3 could be omitted and numbers described in the text. The case numbers per year are too small to allow for a meaningful presentation in a figure.

Author Response

Dear esteemed editor and reviewers,

    We are very grateful of the comments given by the editor and reviewers. The point-to-point correspondence to each comment is listed as follows. We hope the revised manuscript is suitable for publication in Diagnostics.

Sincerely yours,

Tuan-Jen Fang, MD

Chang Gung Memorial Hospital, No. 5 Fushing St., Taoyuan 333, Taiwan

Tel.: +(886)-3328-1200 ext. 3846; Fax: +(886)-3328-1200 ext. 2667;

E-mail: fang3109@cgmh.org.tw

Reviewer 2.

In the abstract, you should also present results supporting your conclusion that "EIL is effective in improving UVFP-related symptoms,". So far, there are none.

Thanks, we add the following sentence in Abstract according to our results.

“After EIL, the glottal gap area, maximum phonation time and voice outcome survey showed significant improvement at 1-, 3- and 6-month measurements.”

In the abstract, you will also need to state that the survival advantage for patients with UVFP is found only in univariate analysis, and vanishes when doing a multivariable analysis. The way you present results now is misleading.

Thank you for the comment. We add the following sentence in Abstract, as:

“Multivariate Cox logistic regression analysis showed two independent predictors of OS , including primary stage and anastomosis type.”

In the introduction, you should already provide a brief description of what EIL actually is, so readers unfamiliar with the technique can get a first idea.

Thanks for the suggestion. We added a brief introduction of EIL as follows:

“EIL is the procedure of injecting the temporary material into the paralyzed vocal fold to correct the vocal position. The voice and swallowing function can be restored immediately after the procedure.”

In the multivariable regression analysis, I suggest defining stage I/II as reference category (OR=1), and stage III/IV as comparator. This would make ORs across possible risk factors more easily comparable. In the methods section, you fail to accurarely describe how you construct and calculate the multivariable model.

We have corrected accordingly.

The following sentences were also added in the statistics section.

“Multivariate analysis of survival was conducted using a stepwise logistic regression model fitted using a backward selection procedure by including variables with P values < 0.05 in univariate analysis.”

The fact that the effect of UVFP on survival loses significance after multivariable adjustment reflects the fact that there is selection who gets radical lymph node dissection along the RLN.

Thanks for the comment. We noted patients of thoracoscopy with cervical anastomosis has higher rate of UVFP. The above factors and numbers of node may also decrease the role of UVFP in OS. We added the following sentences in the discussion section.

“However, after multivariate analysis, its role on survival was diminished. We noted patients of thoracoscopy with cervical anastomosis has higher rate of UVFP. The above factors and numbers of dissected node may also decrease the role of UVFP impaction on OS. The fact that UVFP didn’t has worse outcome can be attributed to the in-troduction of EIL.”

Figure 3 could be omitted and numbers described in the text. The case numbers per year are too small to allow for a meaningful presentation in a figure.

Thanks for the suggestion. Since the time point in performing EIL can only be shown in Figure 3, we would prefer to keep Figure 3

Round 2

Reviewer 1 Report

Thanks for revising your manuscript which is now suitable for publication

This manuscript is a resubmission of an earlier submission. The following is a list of the peer review reports and author responses from that submission.